# Real-Time, Smart Rainwater Storage Systems: Potential Solution to Mitigate Urban Flooding

**Ruijie Liang [1], Michael Di Matteo [2], Holger R. Maier [1],* and Mark A. Thyer [1]**

[1] School of Civil, Environmental and Mining Engineering, University of Adelaide, Adelaide,
  SA 5005, Australia; ruijie.liang@adelaide.edu.au (R.L.); mark.thyer@adelaide.edu.au (M.A.T.)
[2] Water Technology Pty. Ltd., Adelaide, SA 5063, Australia; michael.dimatteo@watertech.com.au
* Correspondence: holger.maier@adelaide.edu.au; Tel.: +61-(0)8-8313-4139

**Abstract:** Urban water systems are being stressed due to the effects of urbanization and climate change. Although household rainwater tanks are primarily used for water supply purposes, they also have the potential to provide flood benefits. However, this potential is limited for critical storms, as they become ineffective once their capacity is exceeded. This limitation can be overcome by controlling tanks as systems during rainfall events, as this can offset the timing of outflow peaks from different tanks. In this paper, the effectiveness of such systems is tested for two tank sizes under a wide range of design rainfall conditions for three Australian cities with different climates. Results show that a generic relationship exists between the ratio of tank:runoff volume and percentage peak flow reduction, irrespective of location and storm characteristics. Smart tank systems are able to reduce peak system outflows by between 35% and 85% for corresponding ranges in tank:runoff volumes of 0.15–0.8. This corresponds to a relative performance improvement on the order of 35% to 50% compared with smart tanks that are not operated in real-time. These results highlight the potential for using household rainwater tanks for mitigating urban flooding, even for extreme events.

**Keywords:** smart rainwater tanks; real-time control; urban flooding; simulation-optimization; genetic algorithms

## 1. Introduction

Urban water supply systems are experiencing unprecedented changes due to population growth [1], increased urbanization [2], and climate change [3]. Population growth and increased urbanization lead to an increase in demand for water resources [4], while climate change is more likely to reduce the amount of water that is available to meet this demand [5]. These are creating a number of challenges for current urban water systems, as well as the design of and planning for future systems.

Household rainwater tanks have been shown to be an effective means of assisting with addressing this problem, as they have the ability to supplement existing water supplies by using a water resource that would otherwise not be utilized. For example, Coombes and Barry [6] reported that household rainwater tanks can significantly increase the resilience of water supply systems under natural variations and future climate change. Similarly, Newman et al. [7] and Burns et al. [8] suggested that tank water usage can lead to a reduction in mains water use, which will help existing water supply systems to meet required demand. Paton et al. [9,10] and Beh et al. [11,12] found that additional supplies from household rainwater tanks, along with those from other sources, such as stormwater harvesting and desalinated water, can form a part of optimal integrated strategies for increasing regional water supply security for cities.

In addition to increasing water supply security, household rainwater tanks have a number of other benefits, such as improving the water quality of receiving waters [13,14] and reducing peak flows for short-duration storm events [15–19]. The ability of household rainwater tanks to reduce peak flows is of particular interest, as not only water supply systems, but also stormwater systems, are likely to be adversely affected by increased urbanization and climate change. Increased urbanization is often associated with urban infill and densification, which will increase the imperviousness of urban catchments, and hence result in increased runoff [20]. In addition, climate change is likely to cause more extreme rainfall events [21], placing further pressure on existing stormwater systems.

However, the capacity of rainwater tanks to reduce discharge rates from connected roofs is generally not fully utilized in practice, as they are commonly not empty during storm events [22]. This limitation can be overcome with the aid of smart technologies, which enable rainwater tanks to be emptied based on knowledge of impending rainfall events, thereby maximizing available retention storage [14,23–25]. For example, South East Water (Melbourne, Australia) use controlled outlets to empty rainwater tanks before a forecasted storm event, which can maximize retention capacity to reduce peak flows [26,27]. However, smart rainwater tanks operated by simply emptying tanks prior to a storm event, so that they essentially behave like a retention tank during a storm event, can be limited in their ability to reduce peak flows for storm events that have large volumes of runoff [18,28]. Consequently, while tanks that are emptied prior to the arrival of storm events with the aid of smart technologies are able to deal with nuisance flooding, they are generally unable to prevent the upgrade of existing stormwater systems to cope with storms associated with the increased runoff resulting from the impacts of urbanization and climate change [18].

In order to address this shortcoming, Di Matteo et al. [29] introduced an approach for controlling the outflow from systems of rainwater tanks in real time during a storm event so as to minimize system peak flow rate based on knowledge of future rainfall patterns. They showed that by using a real-time systems control strategy during the storm, this approach is able to reduce peak flows by up to 48% under conditions that result in large runoff volumes (i.e., 1 in 100 year rainfall event of 24 h duration), compared with no reduction in peak flow when tanks are emptied prior to the rainfall event, but not operated as systems during the rainfall event.

However, Di Matteo et al. [29] only considered a single return period, a single rainfall duration and a single location (Adelaide, SA, Australia). While this provides a proof-of-concept of the approach, it does not provide a comprehensive assessment of the effectiveness of the approach under the range of conditions likely to be experienced for different urban catchments. Consequently, the objective of this paper is to compare the effectiveness of tanks that are emptied prior to the arrival of a storm and then operated as systems in real time during storm events (referred to as "real-time smart systems approach" henceforth) and tanks that are emptied prior to the arrival of storms, but not controlled during storm events, under a range of return periods, storm durations, and tank sizes for locations with different climates.

The remainder of this paper is organized as follows. The methodology used to perform the assessment of the effectiveness of the real-time smart systems approach under different conditions is given in Section 2, followed by details of the case study and experimental methods to which this approach is applied in Section 3. An outline and discussion of the case study results are given in Section 4 and a summary and conclusions are provided in Section 5.

## 2. Real-Time Smart Systems Approach

### 2.1. Conceptual Outline

By controlling a number of smart rainwater tanks as systems during storm events, the timing of the peak flows from the sub-catchments contributing to each tank can be shifted, which will lead to a reduction in the peak discharge rate of the system as a whole. In order to illustrate this concept, a typical two-storage smart rainwater tank system is used (Figure 1). In this system, each tank is fed from roof runoff via a system of gutters and downpipes and the outflow from the tanks feeds into a

drainage system (simplified as a freely draining stormwater pipe in the schematic in Figure 1). Both tanks are emptied prior to the arrival of a storm and the outflow from each tank is controlled independently throughout the storm event via remote-controlled, actuated orifices so as to minimize the total system outflow based on information on the temporal distribution of the incoming rainfall.

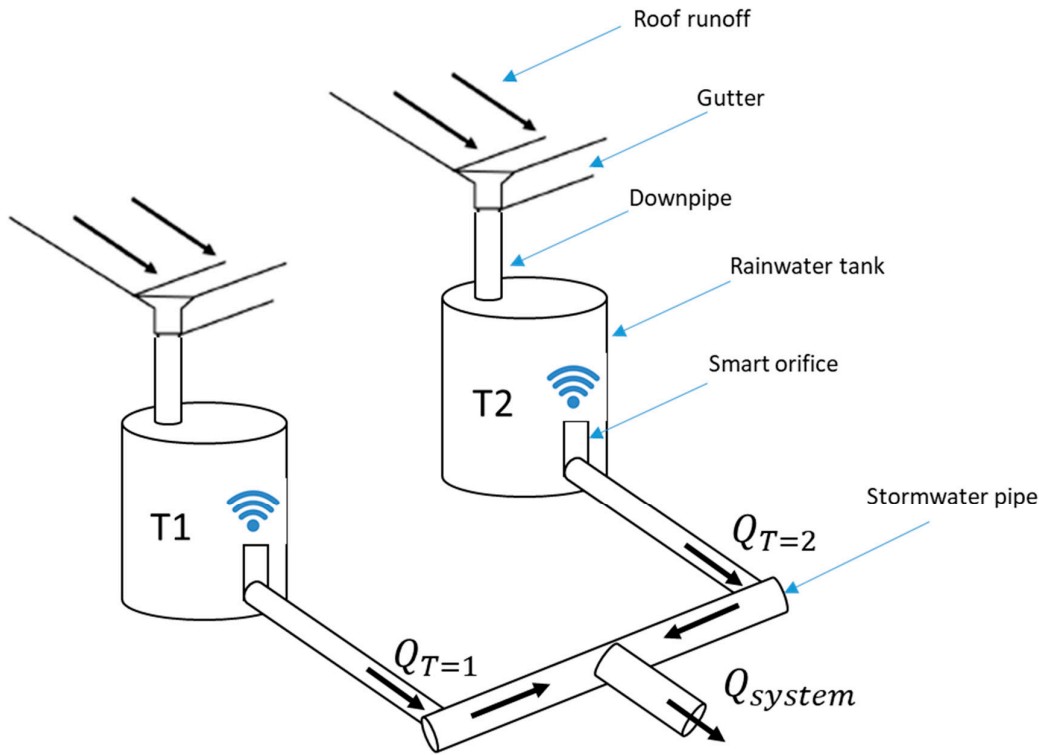

**Figure 1.** Schematic of an example two-tank system operated using the real-time, smart systems approach.

A conceptual representation of how the use of the real-time, smart systems approach is able to reduce peak flows is given in Figure 2. In this figure, the behavior of the real-time smart systems approach is compared with that of a benchmark approach, as part of which the tanks are drained prior to the arrival of the storm, as is the case with the real-time, smart systems approach, but where the orifices remain closed during the rainfall event so that the tanks behave as retention tanks during the storm. As can be seen, when the benchmark approach is used (Figure 2a), both tanks are starting to fill from the beginning of the storm. Once the tanks are full, but rainfall continues, both tanks overflow, and the system operates as though there is no storage from that point onwards. As a result, the peak flows from the two sub-catchments are coincident, resulting in a relatively large outflow from the system as whole.

In contrast, when the real-time smart systems approach is used, the outlet of one of the tanks remains open at the beginning of the rainfall event. As a result, the peak outflows from the two tanks do not occur at the same time, but are distributed over a longer time period, reducing the peak outflow from the system as a whole (Figure 2b). By offsetting the outflows from the two tanks, both the stormwater system and the available storages are being utilized more effectively. With regard to the stormwater system, the real-time smart systems approach enables the system to be used throughout the duration of the entire rainfall event, rather than being idle for part of the rainfall event while the tanks are being filled and then receiving a high load once both tanks spill. With regard to the storages, the real-time smart systems approach enables the available storage to be used at different times, ensuring empty storage is available throughout the rainfall event and enabling the outflow hydrographs from the two sub-catchments to be staggered.

With the help of real-time control, numerous control strategies can be used to reduce the system peak flow rate. Which strategy is optimal is a function of several complex, interacting factors, such as rainfall pattern, time of concentration, tank capacity, etc. Despite this complexity, the principle illustrated above underpins most of these strategies. However, which strategies maximize peak system outflow needs to be determined for particular systems and storm events using advanced optimization techniques. Details of the formulation of the above optimization problem are given in the following sub-section.

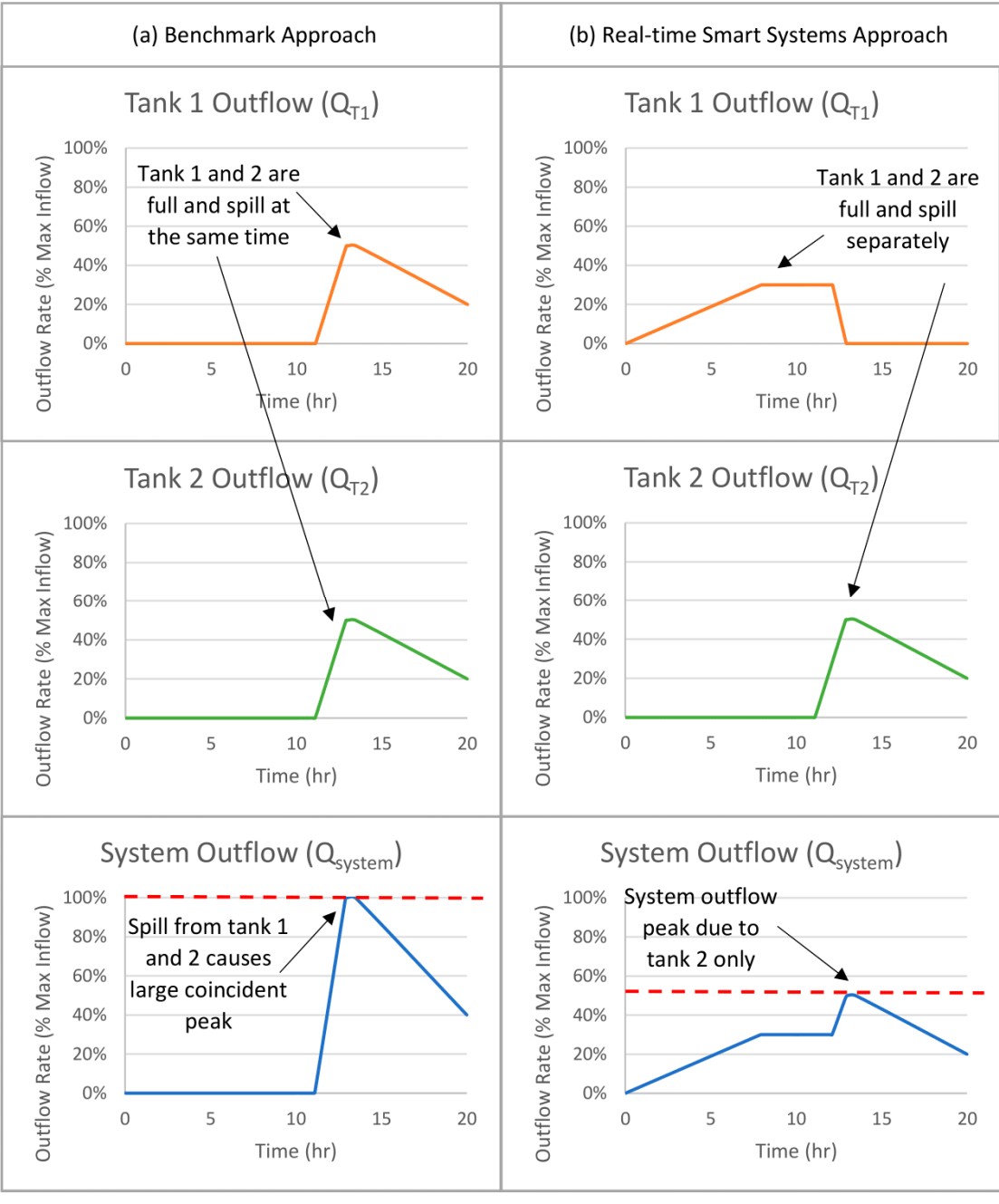

**Figure 2.** Conceptual illustration of performances of (**a**) "benchmark approach" (i.e., tanks emptied prior to the arrival of the storm, but not controlled during the storm) and (**b**) "real-time smart systems approach" (i.e., tanks emptied prior to the arrival of the storm, and controlled as a system during the storm so as to minimize peak system outflow).

## 2.2. Formulation of Optimization Problem

As mentioned above, as part of the real-time smart systems approach, the outlets of systems of tanks can be operated independently during a rainfall event. Given the large number of choices associated with when to open and close each of the tank outlets, and by how much, as well as the variability in rainfall events, it is challenging to identify control schemes that maximize flood peak reduction. Consequently, the optimal control strategies for each tank are identified using a formal optimization approach, based on knowledge of the hyetograph of an incoming rainfall event. It should be noted that these hyetographs are assumed to be known as part of the experiments conducted in this study, as was the case in Di Matteo et al. [29], thereby providing a theoretical upper bound on the effectiveness of the real-time smart systems approach.

In order to enable the peaks of the hydrographs from different roofs to be offset, the outflow from each tank is adjusted by changing the timing and degree of opening of the orifices. Consequently, the decision variables of the formal optimization problem are the percentage opening of the orifice for each tank, ranging from 0% (fully closed) to 100% (fully open), for each control time step during a rainfall event. The number of control time steps and the control horizon depend on the storm duration and number of time steps desired. For this optimization problem, the decision variables for the *i*th control strategy are given as:

$$DV_i = [O_{T=1}^{t=0}, O_{T=1}^{t=1}, \dots, O_{T=1}^{t=N}, \dots, O_{T=S}^{t=0}, O_{T=S}^{t=1}, \dots, O_{T=S}^{t=N}] \tag{1}$$

where, $O_T^t$ is the orifice opening fraction for the *t*th control time step for a control horizon with $N$ time steps, and for tanks $T = 1, 2, \dots S$, where $S$ is the number of tanks being controlled in the system.

The optimization objective is to identify the control scheme(s) that minimize(s) the peak flow rate leaving the system. The objective function of the formal optimization problem is therefore given by:

$$MINIMIZE\{\max(Q_{system})\} \tag{2}$$

where, max($Q_{system}$) is the peak flow rate measured at the system outlet.

## 2.3. Optimization Process

In order to solve the optimization problem outlined in Section 2.2, a simulation-optimization approach is used [30,31] (Figure 3). An evolutionary algorithm [30] is used to select values of the decision variables (i.e., the combination of the degree of opening of the orifices of each tank at each time step) (see Section 2.2). A stormwater simulation model is then used to evaluate the peak flow rate at the system outlet for the selected values of decision variables. Based on the relative success of the selected decision variable values in reducing system peak flow rates, these values will be adjusted using the operators of the evolutionary algorithm (i.e., selection, cross-over, and mutation) so as to further reduce peak flows. This process of selecting a particular control strategy with the aid of the evolutionary algorithm, evaluating the effectiveness of this strategy using a simulation model, adjusting the control strategy based on the relative success of the previous strategies using the evolutionary algorithm, etc. is repeated hundreds or thousands of times until certain stopping criteria have been met, such as completing a fixed number of iterations or until there has been no reduction in peak flows for a certain number of iterations [31].

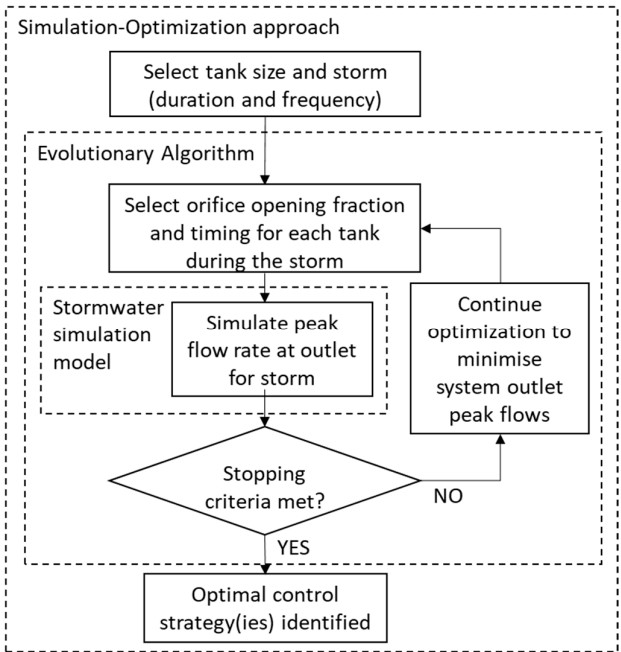

**Figure 3.** Details of the simulation-optimization approach used to identify tank outflow control strategies that minimize system peak outflows.

## 3. Case Study and Experimental Methods

### 3.1. System Configuration

The effectiveness of the real-time smart systems approach is tested for a theoretical residential two-allotment catchment adapted from Di Matteo et al. [29], as illustrated in Figure 1. Details of the configuration of the case study system are given in Table 1. The catchment consists of two 200 m² roofs, each of which is fully connected to a rainwater tank. The outlets of the rainwater tanks are directly connected to the stormwater pipe, which is assumed to discharge freely. It is assumed that there is no initial loss for the storm event, which enables the volumes of runoff from various storms to be directly compared. The tank height is set as 2 m to represent typical above-ground rainwater tanks. As mentioned previously, the orifice opening percentages are the decision variables and are therefore determined with the aid of a genetic algorithm as part of the optimization process. This simple system was selected to enable the impact of the control rules on the ability to reduce system peak flows to be isolated and to enable the results to be applicable to other catchments with different roof sizes.

**Table 1.** Configuration of case study system.

| Design Parameter | Value |
|---|---|
| Orifice opening percentage (%) | Variable |
| Tank height (m) | 2 |
| Roof catchment size (m²) | 200 |
| Percentage of roof connected to tank (%) | 100 |
| Initial loss (mm) | 0 |
| Number of roofs | 2 |
| Number of tanks | 2 |

## 3.2. Implementation of Simulation-Optimization Approach

The simulation-optimization approach (Section 2.3) was implemented by linking two existing software packages in the Python language: DEAP (Distributed Evolutionary Algorithms in Python, developed by the Computer Vision and Systems Laboratory at Université Laval, in Quebec City, Canada. [32]) and PySWMM (Python Wrapper for Stormwater Management Model, developed by EmNet LLC in South Bend, the United States [33]). DEAP (v1.3.0) is an evolutionary computation framework developed for solving real-world problems by applying evolutionary algorithms to simulation modules and is used to select the decision variable values i.e., the degree and timing of the opening of the tank outlets (see Sections 2.2 and 2.3) [34]. The NSGA-II genetic algorithm (Non dominated sorting genetic algorithm) was chosen in the DEAP package, as its variants have already been used successfully for the optimization of urban stormwater systems [29,35,36]. For each optimization run, a population size of 500 was used and the optimization process was continued for 1000 generations to ensure the convergence of the optimization process [29]. All optimization runs were repeated three times from different random starting positions in the decision variable space due to the stochastic nature of genetic algorithms.

The stormwater simulation model used to evaluate the peak flow performance of the controlled tank systems was SWMM (Stormwater Management Model, developed by the United States Environmental Protection Agency [34]). SWMM (v5.1.012) is a widely used dynamic rainfall-runoff-subsurface runoff model that enables the flows from the outlets of rainwater tanks to be controlled with the aid of PySWMM (v0.5.1). As mentioned above, these control schemes are selected by the genetic algorithm implemented in the DEAP package, thereby operationalizing the approach illustrated in Figure 3.

## 3.3. Computational Experiments

In order to test the effectiveness of the real-time smart systems approach, a number of computational experiments were conducted for storm events with different annual exceedance probabilities (AEPs), which correspond to the percentage of a particular storm event being exceeded in any one year, durations and storm patterns in three Australian capital cities (Adelaide, Melbourne and Sydney), as summarized in Table 2. These cities were chosen as their different climates produce different extreme rainfall intensity (see Table 3 for examples of the design rainfall intensity for the 1% AEP). As summarized in Li et al. [37], Adelaide's Mediterranean climate with wet winters and hot dry summers has the lowest rainfall intensity of all three cities (see Table 3). Melbourne's moderate oceanic climate produces severe events in spring and summer due to thunderstorms, resulting in higher rainfall intensity than in Adelaide. Sydney's temperate climate produces the highest rainfall intensity of all three cities, with the extreme rainfall produced in summer, by convective processes.

**Table 2.** Summary of experiment configurations.

| Parameter | Value |
|---|---|
| Location in Australia | Adelaide, South Australia<br>Melbourne, Victoria<br>Sydney, New South Wales |
| Storm frequency (% AEP) | 50, 10, 5, 2, 1 |
| Storm duration | 30 min, 1 h, 6 h, 12 h, 24 h |
| Storm pattern | ten burst patterns |
| Tank size ($m^3$) | 2, 10 |
| Orifice opening percentage (%) | 0% (Fully closed), 10%, …, 90%,<br>100% (Fully open) |
| Orifice diameter | 20 mm |
| Control update time step | 5 min for 30 min, 1 h storms<br>1 h for 6 h, 12 h, 24 h storms |

As the real-time smart systems approach is optimized for a particular rainfall event, its performance can be quite sensitive to the characteristics of this event. Consequently, experiments were repeated for five annual exceedance probabilities, including 1%, 2%, 5%, 10%, and 50%, as well as five different durations, including 30 min, 60 min, 360 min (6 h), 720 min (12 h), and 1440 min (24 h). These were selected as they represent typical ranges of AEPs and critical durations that might be of interest for sub-catchment scale urban drainage infrastructure.

**Table 3.** Design Rainfall Intensity (mm/h) for the 1% annual exceedance probability (AEP) event.

| Duration | 30 min | 1 h | 6 h | 12 h | 24 h |
|---|---|---|---|---|---|
| Adelaide | 67.4 | 43.5 | 12.3 | 7.2 | 4.1 |
| Melbourne | 78.3 | 48.6 | 13.5 | 8.54 | 5.46 |
| Sydney | 118 | 76.7 | 26 | 18 | 12.5 |

Source: Australian Bureau of Meteorology Design Rainfall System [38].

Storm temporal patterns can have a significant impact on design peak flow estimates. Use of a single storm temporal pattern could introduce significant biases in the estimate of the design flow [39]. Consequently, in order to obtain unbiased estimates of peak design flow so that the effectiveness of the real-time smart systems approach can be evaluated in a robust manner, the approach recommended by Australian Rainfall and Runoff 2019 (ARR2019) [39] was adopted. As part of this approach, design peak flows for a given duration and AEP are estimated by taking the average peak flow from ten different storm temporal patterns, instead of using a single storm temporal pattern. These temporal patterns were selected based on the recommendations provided by ARR2019 [39]. Consequently, for the remainder of this paper, the term 'peak flow' for a given duration and AEP refers to the design peak flow estimated from the average of the 10 storm temporal patterns.

All computational experiments were repeated for two tank sizes, including 2 m³ and 10 m³. A 2 m³ tank was considered to be a reasonably popular size for a rainwater tank in Australia, while a 10 m³ rainwater tank was selected as a reasonable upper limit to a publicly acceptable size for residential allotments in an urban infill area.

For the real-time smart system approach, eleven different degrees of opening were considered for each tank outlet, consisting of orifice openings corresponding to 0% (fully closed), 10%, 20%, …, 100% (fully open) open, for an orifice diameter of 20 mm. These openings were implemented for two different control time steps, depending on storm duration. As can be seen from Table 2, a 5 min control time step was used for storms of 30 and 60 min duration, whereas a 1 h time step was used for storms of 6, 12 and 24 h duration. This was done to strike an appropriate balance between search space size and the ability to identify the control strategy that minimizes system peak outflow.

*3.4. Performance Assessment*

In order to assess the effectiveness of the real time, smart systems approach, its performance was compared to that of the benchmark approach, as defined in Section 2.1. For both these approaches, the tanks were assumed to be empty prior to the start of the rainfall event. As mentioned previously, the key difference between the two approaches is that for the benchmark approach, tank outflows are not controlled during the rainfall event, with the orifice remaining closed, whereas for the real-time smart systems approach, the orifice opening/closing of each tank is optimized independently during the rainfall event so as to maximize system peak flow reduction, as explained in Section 3.

To evaluate the performance of both of these approaches, a baseline scenario with no tanks was chosen as a basis of comparison. Therefore, the peak flow rate reduction for the $i^{th}$ experiment configuration and $j^{th}$ storm event is given by:

$$\text{System peak flow reduction } = \left(1 - \frac{\max\left(Q_{Max}^{i,j}\right)}{\max\left(Q_{Max}^{baseline,j}\right)}\right) \times 100(\%), \tag{3}$$

where, $\max\left(Q_{Max}^{i,j}\right)$ is the peak flow of one specific trial, and $\max\left(Q_{Max}^{baseline,j}\right)$ is the peak flow from the case without a tank ("No tank").

## 4. Results and Discussion

### 4.1. Performance of Real-Time, Smart System Approach

The results for the two-tank case study considered show that the real-time smart systems approach can be highly effective in reducing peak flows in urban stormwater systems, with minimum peak flow reductions of ~30% under even the most severe rainfall conditions considered, provided the temporal variation of the incoming rainfall event is known (dashed orange lines, Figure 4). For Adelaide and Melbourne, this level of performance can be achieved using a 2 m³ tank (dashed orange lines, Figure 4a,c, respectively), whereas for Sydney, this requires a 10 m³ tank (dashed orange line, Figure 4f), as a result of the higher intensity rainfall, and hence higher runoff volumes, experienced in this city. For less severe events, such as those with an AEP of 10%, use of the real-time, smart systems approach is able to achieve even greater minimum peak flow reductions of around 60% for 10 m³ tanks. (dashed orange lines, Figure 4b,d,f).

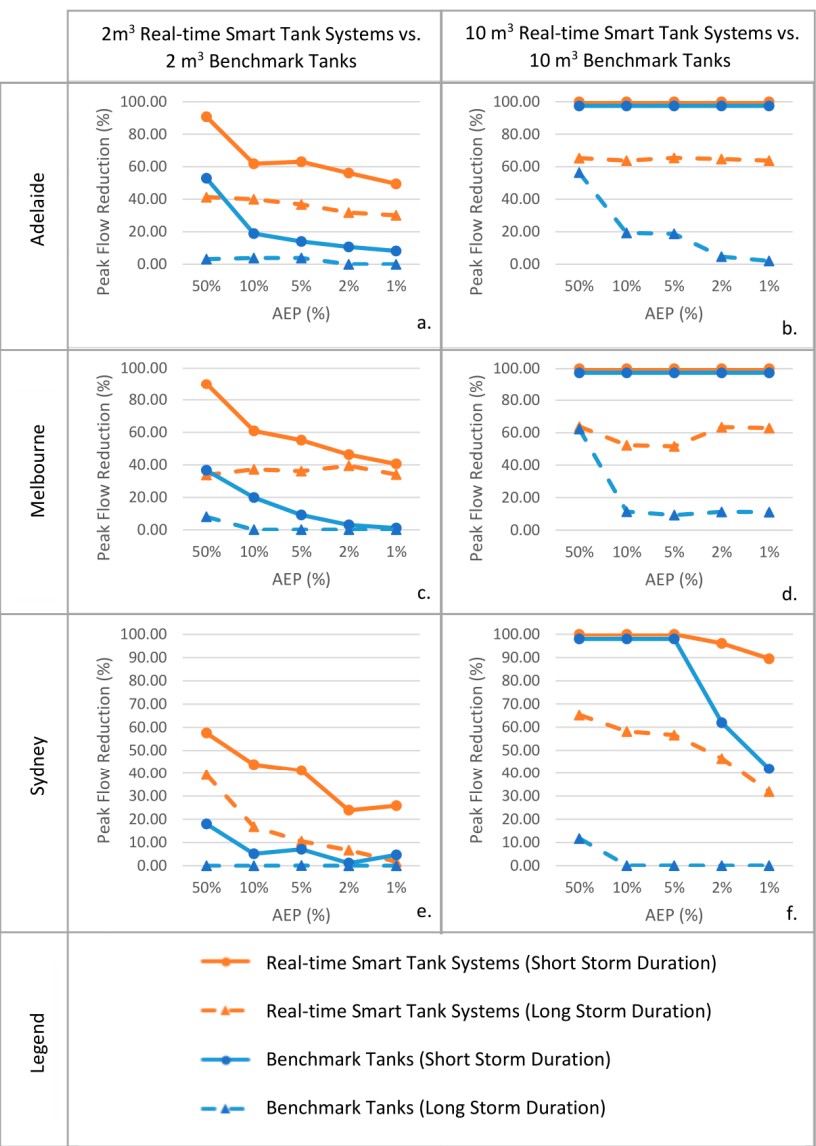

**Figure 4.** Performance of real-time, smart tank systems versus benchmark tanks for the three locations (Adelaide, Melbourne, and Sydney), two tank sizes (2 × 2 m³ and 2 × 10 m³), and five AEPs (50%, 10%, 5%, 2%, and 1%) considered. For the sake of clarity, only results for the shortest (30 min) and longest (24 h) durations considered are shown, with results for the full set of durations considered shown in

Figure 5 and Appendix A. The figure shows the performance of 2 m³ real-time smart tank systems versus benchmark tanks for (**a**) Adelaide, (**c**) Melbourne (**e**). Sydney and 10 m³ real-time smart tank systems versus benchmark tanks for (**b**). Adelaide, (**d**). Melbourne (**f**) Sydney.

The results in Figure 4 also show that long-duration events (24 h—solid orange lines) are more critical than short duration events (30 min—dashed orange lines) in terms of the ability of household rainwater tanks to reduce flood peaks, with the peak flow reductions obtained for the shorter duration events generally on the order of 20% and 40% greater than those obtained for the corresponding longer duration event for the 2 m³ (Figure 4a,c,e) and 10 m³ (Figure 4b,d,f) tanks, respectively. This trend in the decreasing effectiveness of household rainwater tanks in reducing flood peaks for longer duration events is confirmed by the results obtained for the intermediate durations (see Figure 5 for results for Adelaide for 2 m³ tanks and Appendix A for similar results for other locations and tanks sizes) and is in agreement with the findings and assumptions in previous studies [18,29]. It should be noted that although shorter duration events generally result in larger peak flows for catchments without tanks, this is generally not the case once tanks have been added. This is because the runoff volume produced by shorter duration events can generally be fully contained within the tanks. In contrast, while the intensity of long-duration events is less, the runoff volume produced is larger, often exceeding the capacity of the tanks. As a result, for catchments with tanks, and downstream detention infrastructure operating near capacity, attenuation of runoff for longer duration events is generally more critical, as these events produce potentially significant peak flows leaving the catchment (post-tank, as can be seen in Appendix B).

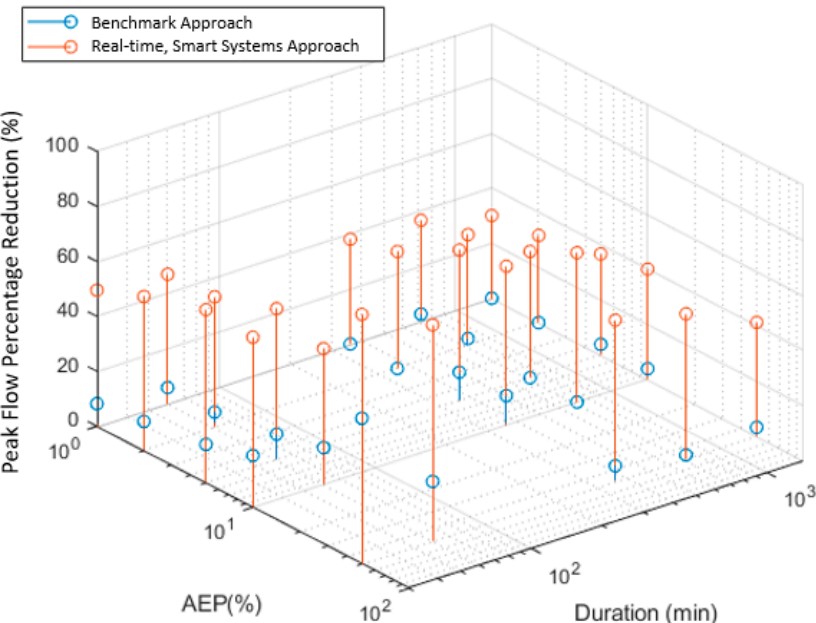

**Figure 5.** Percentage peak flow reduction of benchmark tanks and real-time smart tank systems for a range of durations and AEPs with 2 × 2 m³ tanks for Adelaide.

In general, the performance of the real-time smart systems approach (orange lines) is noticeably better than that of the benchmark approach (blue lines) for the same rainfall duration (e.g., either dashed or solid lines) (Figure 4, Figure 5 and Appendix A). As can be seen in Figure 4, Exceptions are:

(i)   When the available tank storage exceeds the total rainfall volume, as is the case for the short-duration rainfall events for the 10 m³ tanks for Adelaide (Figure 4b), Melbourne (Figure 4d) and AEPs of 50%, 10%, and 5% for Sydney (Figure 4e), where both approaches result in 100% peak

flow reduction (i.e., the solid orange and blue lines are both at 100%), as all of the runoff is able to be retained in the tanks.

(ii) When the available tank storage is only slightly less than the total rainfall volume, in which case the benchmark storage still performs well, as is the case for an AEP of 50% for the long duration events for the 10 m³ tank for Adelaide (Figure 4b) and Melbourne (Figure 4d) (i.e., the dashed orange and blue lines are close together).

(iii) When long duration, extreme events at locations with higher rainfall intensity such as Sydney, are combined with smaller tank volumes (Figure 4e, dashed orange and blue lines), suggesting that the capacity of the tanks is insufficient to mitigate the large volume of runoff generated, even with the real-time smart systems approach.

The above results suggest that the performance of both benchmark tanks and real-time smart tank systems is affected by the volume of runoff generated by a rainfall event, which is a function of location, AEP and rainfall duration, as well as available tank volume. Consequently, in order to enable the above results to be generalized, the ratio of tank volume to total runoff is plotted against peak flow percentage reduction for all computational experiments conducted (Figure 6). As can be seen, the performance of benchmark tanks deteriorates rapidly once the tank volume to runoff ratio drops below 1, to the point where (blue circles and dashed line, Figure 6):

(i)　below ratios of 0.8, peak flow reduction generally drops to below 30%;
(ii)　below ratios of 0.6, peak flow reduction generally drops to below 20%;
(iii)　below ratios of 0.3, peak flow reduction generally drops to below 10%; and
(iv)　below ratios of 0.15, peak flow reduction is generally 0%.

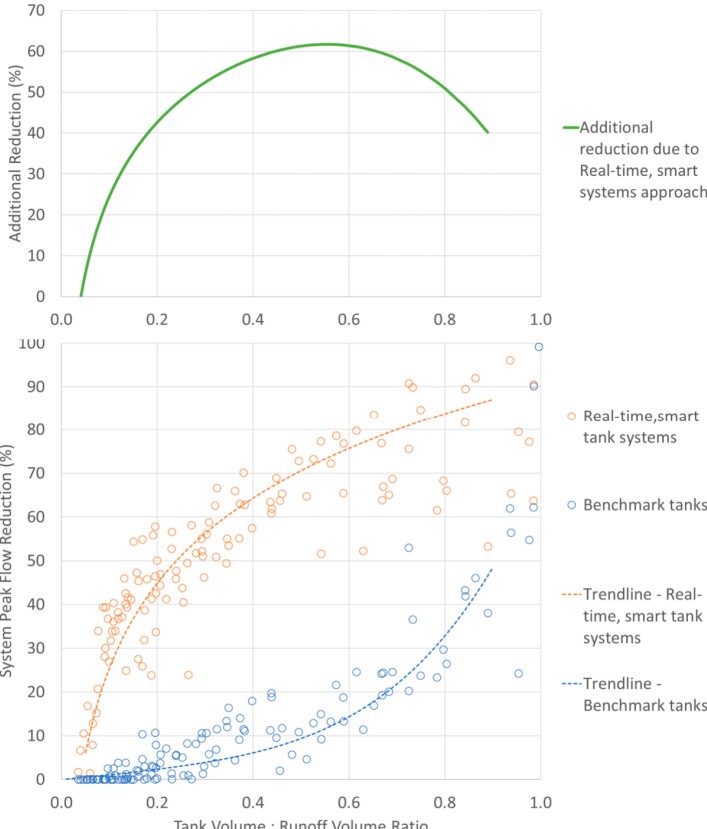

**Figure 6.** Relationship between the ratio of tank to runoff volume and peak flow reduction for all computational experiments (i.e., for all locations, AEPs, rainfall durations, and tank sizes considered) for benchmark tanks (blue circles and dashed line) and real-time smart tank systems (orange circles and dashed line). The green solid line represents the additional peak flow reduction that can be

achieved by using real-time smart tank systems control based on the difference between the two trendlines fitted to the data sets. The trendlines were developed by trial and error and visual inspection for data where $x < 0.9$ (Trendline—Real-time, smart tank systems: $y = 2 \times \ln(x) + 90$ ($r^2 = 0.7953$) and Trendline—Benchmark tanks: $y = 70\,x^4 - 20\,x^3 + 10\,x^2 + 10\,x$ ($r^2 = 0.8327$). Results for tank to runoff volume ratios in excess of 1 are not shown, as they all result in peak flow reductions of 100%.

In contrast, use of the real-time smart systems approach is able to maintain much higher levels of peak flow reduction as the tank volume to runoff volume ratios decrease, where a ratio >0.1 generally results in a minimum peak flow reduction of 30%, which increases to more than 50% for ratios in excess of 0.3 and to more than 60% for ratios in excess of 0.5 (orange circles and dashed line, Figure 6).

For tank to runoff volume ratios between approximately 0.15 and 0.8, the additional peak flow reduction that can be achieved by using the real-time smart system approach, compared with using the benchmark approach, is in the order of 35–50% (green solid line, Figure 6). For ratios greater than this, the benefit of using the real-time smart system approach reduces, not because of a reduction in the performance of these systems, but because of the significantly increased performance of the benchmark approach as the available tank volume approaches the runoff volume generated. For ratios less than 0.15, the performance of the real-time smart system approach deteriorates rapidly, as storage that is less than 15% of the runoff volume is insufficient to balance the outflows from the two tanks in a way that is able to offset their peaks.

In summary, the results of the computational experiments conducted suggest that significant peak flow reductions can be achieved by using the real-time smart systems approach under a wide range of conditions, including extremes. There is a defined relationship between the percentage of peak flow reduction achieved and the ratio of tank to runoff volume for both the benchmark (blue dashed line, Figure 6) and real-time smart tank systems (orange dashed line, Figure 6) approaches. These relationships indicate that while the peak flow reduction ability of the benchmark approach deteriorates rapidly as the ratio of tank to runoff volume decreases, this is not the case for tanks controlled as systems in real-time. In fact, the latter are able to achieve peak flow reductions of 50% with tank volumes that are less than 40% of the runoff volume and only require tank volumes of 20% to achieve peak flow reductions in excess of 40%. In relative terms, use of the real-time smart systems approach is able to achieve between 35% and 60% greater peak flow reductions than use of the benchmark approach for the majority of ratios of tank to runoff volumes (green solid line, Figure 6). This highlights the potential of using real-time, smart tank systems for reducing urban flooding or prevent the need to upgrade existing stormwater systems in certain circumstances.

*4.2. Reasons for Increased Performance of Real-Time Smart Systems Approach*

The way the real-time control of systems of tanks is able to result in significantly greater peak flow reductions than the use of benchmark tanks is illustrated in Figure 7 for a long duration (24 h) event with an AEP of 1% for Adelaide.

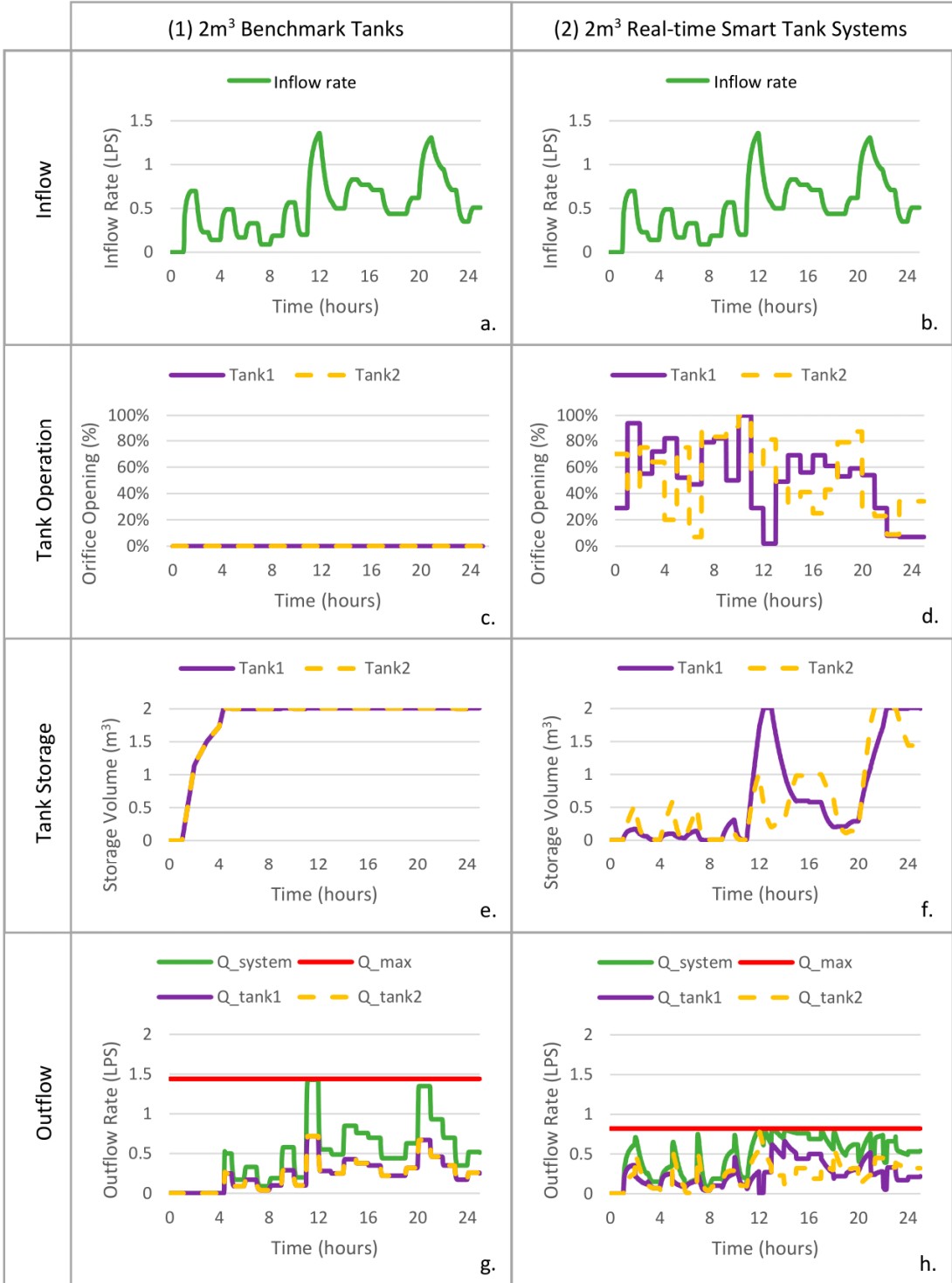

**Figure 7.** Typical operation rules for (**2**) 2 $m^3$ real-time smart tank systems compared to (**1**) 2 $m^3$ benchmark tanks (Adelaide, 24 h duration, 1%AEP).

Figure 7a,b shows that for the benchmark tanks, both orifices are closed throughout the rainfall event (Figure 7c). Consequently, both tanks fill to capacity during the early stages (t = 5 h) of the storm event, which leads to a lack of ability to accommodate the first (t = 12 h) and second (t = 22 h) peaks (Figure 7a,e). As a result, the benchmark tanks spill for the remainder of the rainfall event and have no further influence on outflow (Figure 7e,g). In contrast, when the real-time smart tank systems

approach is used, the orifices of both storages are mostly open before the arrival of the first rainfall peak, which occurs from t = 1 h to t = 11 h (Figure 7b,d), providing sufficient storage capacity to capture this peak and reduce the system outflow (t = 12 h). This is achieved by partially closing the orifices of both tanks (80% closed for Storage 1, and 40% closed for Tank 2).

After the first peak has passed, Tank 1 is emptied (t = 12–13 h), followed by Tank 2, once all of the water has been drained from Tank 1 (t = 13 h) (Figure 7b,f). Before the arrival of the second peak, the orifices of both tanks are almost open to ensure available storage capacity is maximized. Once the second peak arrives (t = 20 h), the orifice of Tank 2 is closed, while the orifice of Tank 1 is closed one time step later at t = 21 h. Both tanks are nearly full (t = 23 h) when Tank 2 is starting to be drained, while the orifice of Tank 1 remains closed. This staggers the outflows from the tanks, enabling the system peak outflow to be reduced from 1.49 L/s when benchmark tanks are used (Figure 7g) to 0.78 L/s when real-time smart tank system controls are used, corresponding to a 48% reduction (Figure 7h).

However, as discussed in Section 4.1, real-time smart systems do not perform markedly better than benchmark tanks when the tank to runoff volume is less than 0.15, as there is insufficient storage to offset peak flows. An example of this is shown in Figure 8 for a long duration (24 h) event with an AEP of 1% for Sydney, for which the storage to runoff volume is 0.034. The reason the real-time smart systems approach achieves less than 5% reduction in this instance is due to a combination of the higher rainfall intensity and the temporal pattern, producing an inflow rate that fills both tanks within the control update time step of 1 h (see Table 2).

During the period t = 4–11 h, the orifices of both tanks remain open (Figure 8b) to ensure the tank is empty prior to the arrival of the main peak of the inflow (Figure 8a). However, the much higher inflow rate that starts around t = 12 h of greater than 3–4 LPS (liter per second) means the 2 m³ tanks become full in the t = 11–12 h period (Figure 8c). An inflow rate of 3 LPS produces an inflow volume of 3.6 m³ in 1 h. This is close to the capacity of the 2 × 2 m³ tank. As the capacity of the tanks is exceeded in a 1 h period, there is limited opportunity to delay the peak using either of the tanks. Hence, the peak of the outflow is similar to the peak of the inflow. A larger storage size is able to solve this problem, so that the volume produced by a high inflow rate during a control update time step can be contained in one of the tanks.

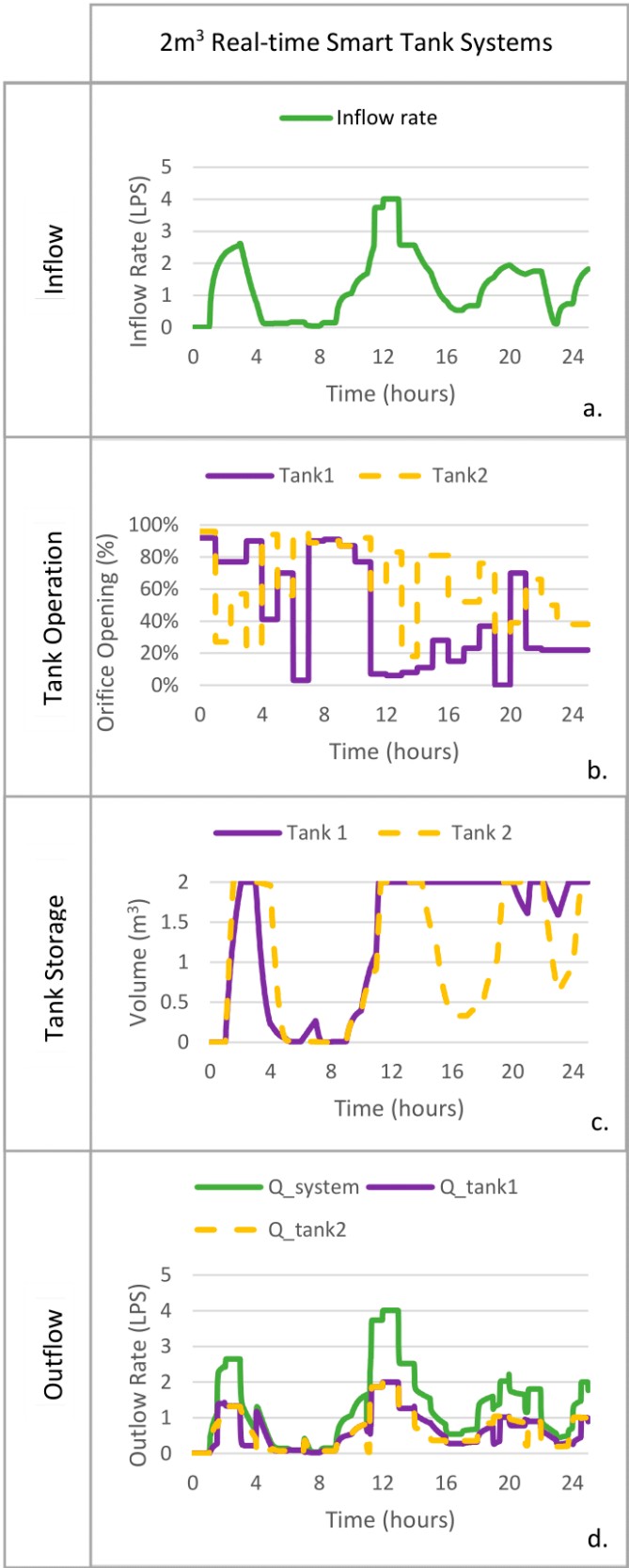

**Figure 8.** Typical operation rules for real-time smart tank systems approach for systems where the ratio of tank:runoff volume is less than 0.15 (Sydney, 24 h duration, 1% AEP).

## 5. Conclusions

This study demonstrates that smart rainwater tanks operated as a system in real-time during a storm event, an approach proposed in Di Matteo et al. [29], can significantly reduce the downstream peak runoff flow rate for a wide range of storm durations (30 min to 24 h) and frequencies (50% to 1% AEP). To the authors' knowledge, this is the first study to demonstrate that household-scale rainwater tanks could potentially provide peak flow attenuation performance across a wide range of storm event durations for rare events (i.e., 10% to 1% AEP). Importantly, the real-time smart systems approach can provide harvesting and other benefits during non-critical periods using the same controlled outlet infrastructure.

In the study, an optimization-simulation model was developed to identify the optimal control strategies that maximize the peak flow reduction performance of a simple two-tank system. The system consisted of $2 \times 200$ m² roof catchments, each connected to a smart rainwater tank with the tank outlets discharging at one point. Two tank sizes, 2 m³ and 10 m³, were tested for design storm events for three Australian cities: Adelaide, Melbourne, and Sydney. The optimal peak flow reduction performance for the real-time smart tank systems approach was compared with the performance of a benchmark approach, where the tanks are emptied prior to a storm event, but behave as retention tanks during the storm.

The results demonstrate that the real-time smart systems approach has significant potential to provide a mitigation option for urban flooding problems that require peak flow rate attenuation at the allotment scale. The results showed that by applying an optimal control strategy for each tank, during a storm event, the tanks could provide from 35% to 85% peak flow attenuation of runoff from the roof catchments connected to the tanks where the tank volume to runoff volume ratios range from 0.15 to 0.8, respectively. This performance represents a peak flow reduction improvement in the order of 35% to 50% compared with the benchmark approach. For tank:runoff volume ratios outside of this range, the relative advantage of the real-time smart systems approach declines, as for ratios >0.8, the benchmark approach also performs well, and for ratios <0.15, the performance of real-time smart tank systems approach reduces significantly.

The underlying reason for the high peak flow reductions achieved using the real-time smart systems approach is that operating tanks as a system provides the potential to apply a control strategy such that water can be released in a way where: (1) there is enough capacity in the tanks to detain peaks in runoff inflows, (2) release of stored water from multiple tanks is staged so as to reduce the occurrence of coincident runoff peaks leaving the system outlet, and (3) there is sufficient capacity to detain subsequent runoff inflow peaks if needed. The critical component of the optimal control strategies that enable this to be achieved is the timing of the opening/closing of the orifices to control tank outflows.

The systems model tested was based on several assumptions in order to identify an upper limit to the theoretical performance of the real-time smart systems approach. These assumptions included: 1) perfect knowledge of incoming storm characteristics prior to the event, 2) accurate orifice control, 3) a simple two-allotment system, and 4) 100% roof to tank connection with no losses (e.g., spill from gutters). Consequently, further research is required before this approach can be applied in practice. Such research should assess the sensitivity of the peak flow reductions achieved in this study to the above assumptions, as well as the applicability of this approach to larger, street-scale systems, especially when integrated with existing stormwater systems. In addition, further technological advances are required in order to be able to reliably control the tank outlets in real time. However, while the proposed approach does not provide a ready-made alternative to more traditional solutions at present, the rapid advances in rainfall forecasting and smart technologies suggest that this approach could provide a viable alternative for reducing flood peaks in urban areas in the not-too-distant future.

**Supplementary Materials:** Three supplementary materials are submitted alongside the manuscript: File S1 Summary peak flow reduction data, File S2 Example operation rules data, and File S3 Example SWMM model.

**Author Contributions:** All authors conceived and designed the experiments. R.L. performed the experiments and all authors analyzed the data. All authors contributed to write the paper.

**Funding:** This research received no external funding.

**Acknowledgments:** The authors would like to thank the Environment Institute and the University of Adelaide for funding this research, Graeme Dandy for contributing to the concept, and Water Technology Pty. Ltd. for providing technical suggestions. Ruijie Liang received a scholarship provided by the University of Adelaide.

**Conflicts of Interest:** The authors declare no conflict of interest.

**Appendix A**

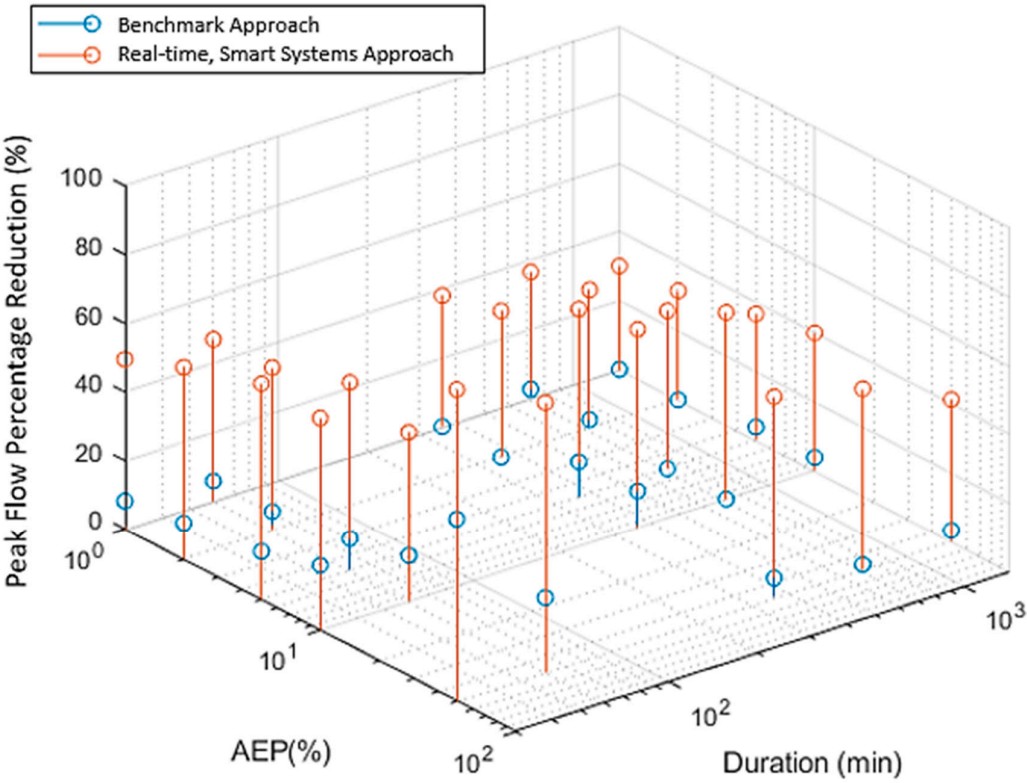

**Figure A1.** Percentage peak flow reduction of benchmark tanks and real-time, smart tank systems for a range of durations and AEPs with $2 \times 2$ m$^3$ tanks in Adelaide.

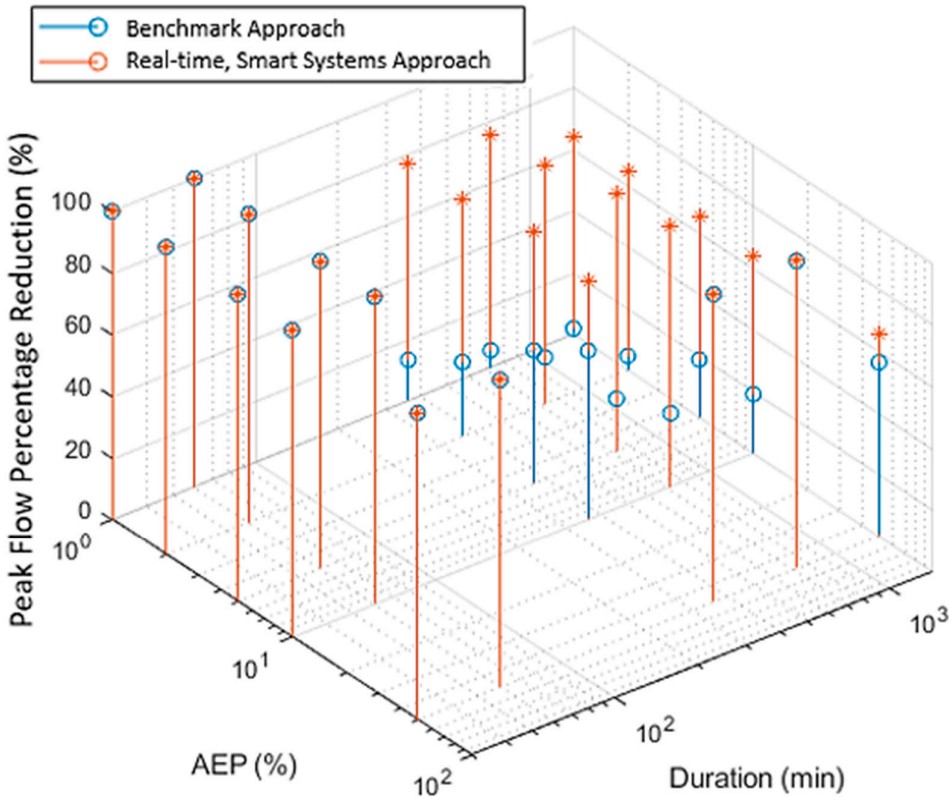

**Figure A2.** Percentage peak flow reduction of benchmark tanks and real-time, smart tank systems for a range of durations and AEPs with 2 × 10 m³ tanks in Adelaide.

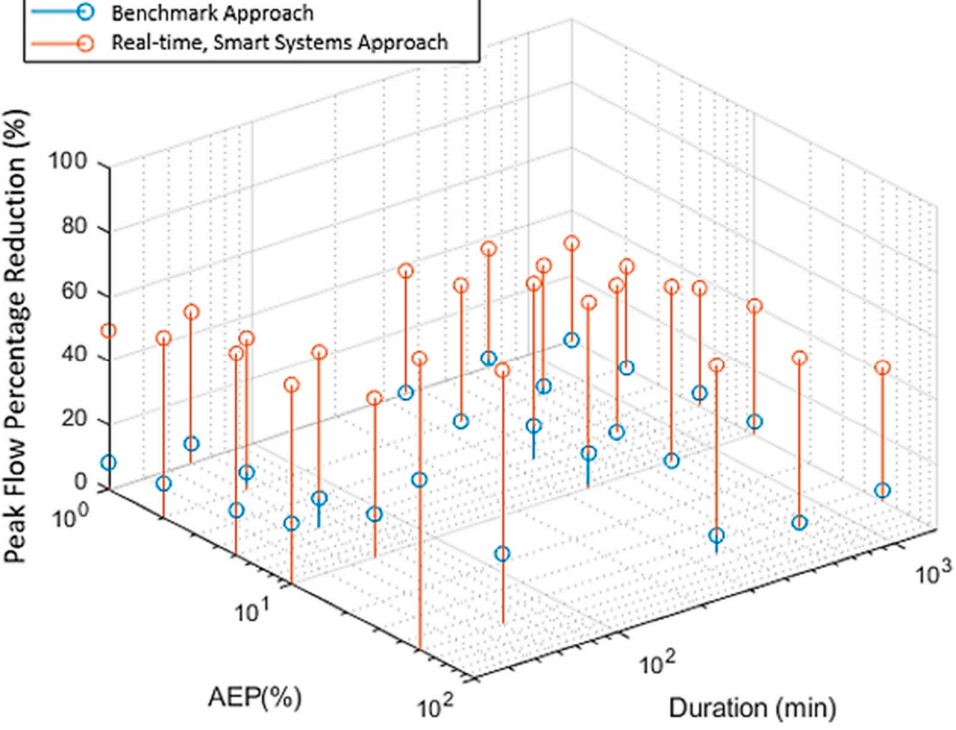

**Figure A3.** Percentage peak flow reduction of benchmark tanks and real-time, smart tank systems for a range of durations and AEPs with 2 × 2 m³ tanks in Melbourne.

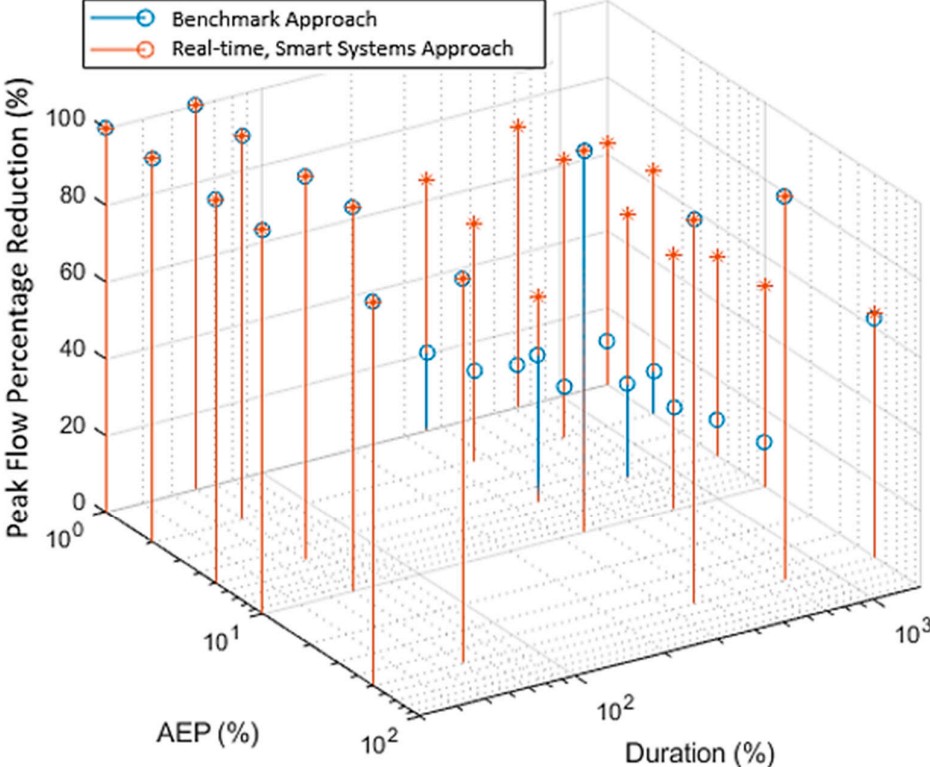

**Figure A4.** Percentage peak flow reduction of benchmark tanks and real-time, smart tank systems for a range of durations and AEPs with 2 × 10 m³ tanks in Melbourne.

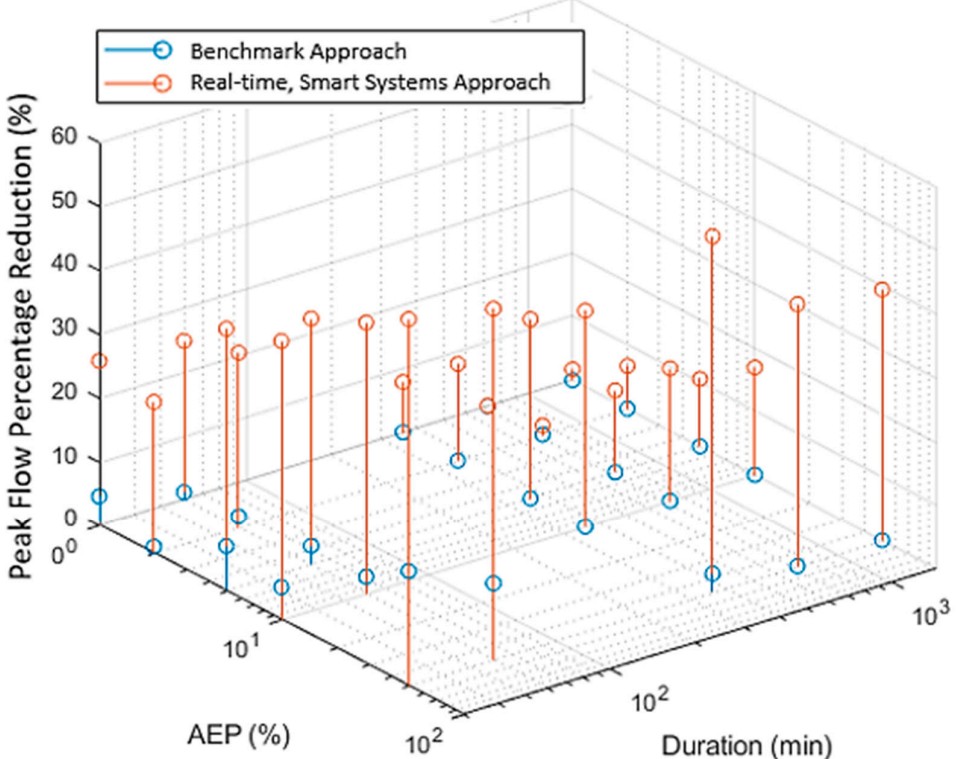

**Figure A5.** Percentage peak flow reduction of benchmark tanks and real-time, smart tank systems for a range of durations and AEPs with 2 × 2 m³ tanks in Sydney.

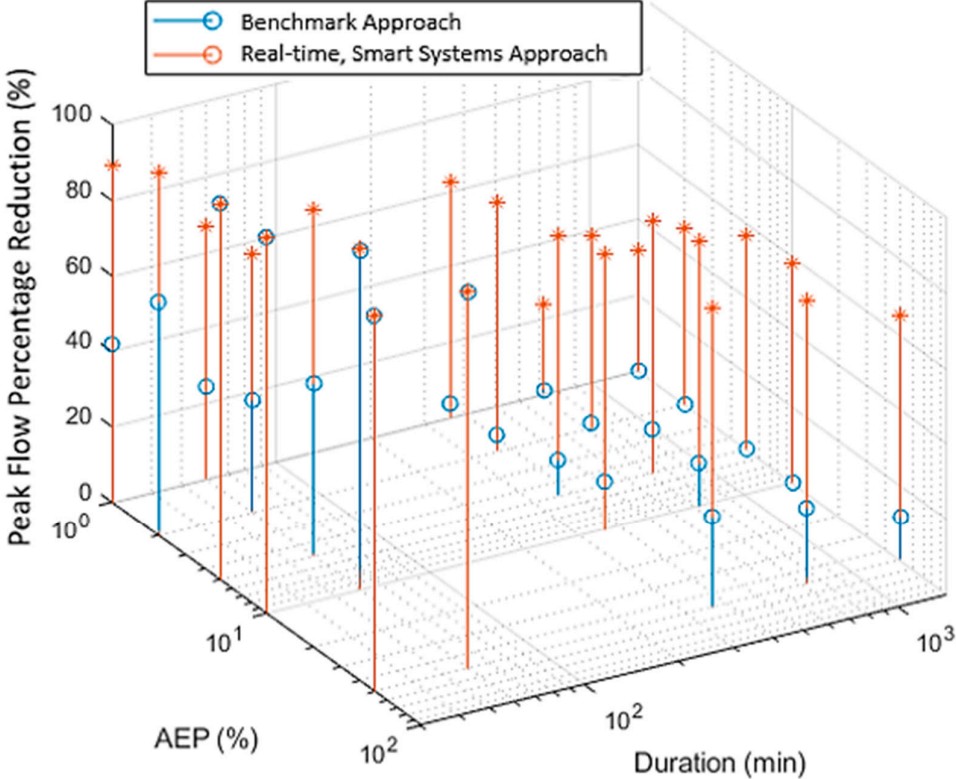

**Figure A6.** Percentage peak flow reduction of benchmark tanks and real-time, smart tank systems for a range of durations and AEPs with 2 × 10 m$^3$ tanks in Sydney.

## Appendix B

**Table A1.** Critical duration of no tank system, benchmark tanks and real-time, smart tank systems for a range of AEPs with 2 × 2 m$^3$ tanks in Adelaide, Melbourne and Sydney.

| Climate | AEP (%) | No Tank System | Benchmark Tanks | Real-Time, Smart tank Systems |
|---------|---------|----------------|-----------------|-------------------------------|
| Adelaide | 1 | 30 min | 30 min | 30 min |
| | 2 | 30 min | 30 min | 30 min |
| | 5 | 30 min | 30 min | 30 min |
| | 10 | 30 min | 30 min | 60 min |
| | 50 | 30 min | 60 min | 60 min |
| Melbourne | 1 | 30 min | 30 min | 30 min |
| | 2 | 30 min | 30 min | 30 min |
| | 5 | 30 min | 30 min | 30 min |
| | 10 | 30 min | 30 min | 30 min |
| | 50 | 30 min | 60 min | 60 min |
| Sydney | 1 | 30 min | 30 min | 30 min |
| | 2 | 30 min | 30 min | 30 min |
| | 5 | 30 min | 30 min | 30 min |
| | 10 | 30 min | 30 min | 30 min |
| | 50 | 30 min | 30 min | 30 min |

**Table A2.** Critical duration of no tank system, benchmark tanks and real-time, smart tank systems for a range of AEPs with 2 × 10 m$^3$ tanks in Adelaide, Melbourne and Sydney.

| Climate | AEP (%) | No Tank System | Benchmark Tanks | Real-Time, Smart Tank Systems |
|---|---|---|---|---|
| | 1 | 30 min | 6 h | 6 h |
| | 2 | 30 min | 6 h | 6 h |
| Adelaide | 5 | 30 min | 12 h | 6 h |
| | 10 | 30 min | 12 h | 6 h |
| | 50 | 30 min | 24 h | 24 h |
| | 1 | 30 min | 6 h | 6 h |
| | 2 | 30 min | 6 h | 6 h |
| Melbourne | 5 | 30 min | 6 h | 6 h |
| | 10 | 30 min | 12 h | 12 h |
| | 50 | 30 min | 24 h | 24 h |
| | 1 | 30 min | 60 min | 60 min |
| | 2 | 30 min | 60 min | 60 min |
| Sydney | 5 | 30 min | 60 min | 6 h |
| | 10 | 30 min | 6 h | 6 h |
| | 50 | 30 min | 6 h | 6 h |

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
