# Peer review of "Real-Time, Smart Rainwater Storage Systems: Potential Solution to Mitigate Urban Flooding"

_water, doi:10.3390/w11122428_

Round 1

Reviewer 1 Report

Dear Mr. Liang and co-authors

Thank you for this very well written article, it was a pleasure to read. I appreciate your efforts towards sharing your work in such a well-structured, well-formulated and well-illustrated manner. The study behind also seems well structured and rigorous, using state-of-the-art methods. I am not an expert in real-time control, so I cannot evaluate whether your choice of optimization algorithm is optimal, but I assess that the overall procedure is sound.

I have a minor problem with some terms you use, which I am guessing must be the norm in Australia but are not familiar to me (in Europe):

“Annual exceedance probability” gets significantly less hits in a simple google search compared to “return period” (usually denoted T) – I suggest that you consider switching to T, or at least explain how AEP relates to T when you introduce AEP in the text. Respecting the metric system, I believe that m3 is the correct way to represent what you denote as kL. Why do you “purge” a rainwater tank rather than simply “empty” it? According to the Cambridge dictionary “to purge” means “to make someone or something free of something evil or harmful” or “to remove something bad or wrong” or “to get rid of people from an organization because you do not agree with them”, all which seems a little too judgmental when it comes to water in a tank…

I have a more serious problem with your setting of your results in a larger perspective. In the two last paragraphs of the conclusions, you do mention all the factors that would affect the significance of the effect of real-time control of rainwater tanks on larger, real-world systems, and state clearly that this study serves to “identify an upper limit to the theoretical performance”. However, you abstain from offering the reader any idea of the order of magnitude of effect that the real-time controlled rainwater tanks would have on real-life systems. This could be because you want to be prudent and would rather wait until you have made simulations to confirm any ideas you may have. The problem is that a reader who is not well versed in the field of urban stormwater management may be lead to believe that real-time control of rainwater tanks is a viable solution to urban flooding, which, I am sure you would agree with me, it is NOT. It may offer a little contribution towards reducing flood extents, but MUST be combined with other measures in order to have any significant effect on a scale larger than a couple of allotments. Please remember that this is an open-access journal and your research is publicly available to anyone, scientists and laymen alike; I believe that it is your obligation to offer a few words of caution about interpretation of the results in order to avoid misunderstandings.

I value that you append supplementary materials such as exemplary results and model set-up. To even better live up to the ideal of open science I would encourage you to also share an exemplary python script.

Below is a list of smaller comments to specific places in the text.

Line 40: I think you are missing the word “of” between “part” and “optimal”

Line 43: I would specific water quality of receving waters (as it is written now it could be misunderstood as quality of the water in the tank or the sewer system)

Line 65-66: there is something missing or needing deletion here

Line 173-174: I do not understand the argument

Line 239: I would use “and” instead of “but”

Line 248-249: you write “minimum peak flow reductions of 30% (averaged over 10 patterns)” – this minimum is thus not the absolute minimum. It would be relevant to know the variation among the results that stems from using the 10 storm patterns (now that you have gone through the pains of including this relevant variable). I encourage you to present in in some way, e.g. a separate graph, or an error bar (error band) on an y of the existing figures.

Line 268-272: You talk about a trend of decreasing effectiveness for longer duration events. But aren’t the peak flows during longer durations generally smaller? And isn’t the problem with flooding due to long duration events generally more related to the overall volumes rather than peak flows that exceed the conveyance ability of sewers? In such case the ability to reduce peak flows during long duration events is less relevant for solving real-world flooding problems, which would be relevant to include in the discussion.

Line 406. See line 239. 

Reviewer 2 Report

General comments: The paper gives an original study to demonstrate that also household scale rainwater tanks could potentially provide peak flow attenuation performance across a wide range of storm durations for rare events. These rainwater management techniques have great potential to contribute achieve them not only in the studied region.

Strengths of paper: The study based on modern tools for three Australian cities with flood control techniques could be a useful and effective part guiding the city closer to a sustainable sponge and resilient city everywhere.

Weakness of paper and suggestion for improvement:

Only a few comments - I do not know abbreviastion kL for tank size. Second one in the abstract missing the unit for runoff volumes.

Reviewer 3 Report

The following minor edits [M] / suggestions [S] and queries [Q} to the authors are defined by line number and titled M, S or Q accordingly to support a final journal-ready version. In the round the paper is sound, well written and begins the journey towards what will doubtless become a series of research works around the development of these concepts. Ideally such a paper would take a larger step (i.e. investigate this at a street-scale) in its first form, however it is acknowledged that this will likely be a “next step”.

43-Q what water quality is improved?

57-M “are improved” – can be?

66 – M reword during storm

90 – Q is it control throughout the storm event… or also before AND After it?

95 – M rephrase

96 – Q is “traditional controls” most appropriate term here?... aren’t all active drain down systems are “novel” in relation to the traditional orifice plate / Vortex flow control option.

121 – Q the authors describe the charts  under (1) Traditional retention approach. Might we be provided a figure / reference that defines how this compares to Figure 1…? Secondly… the traditional retention approach would in my mind be one where the tank has a starting water level somewhere between 0 and 100% (not empty). Where the tank is 100% empty at the start of the storm, I would again urge against using the term traditional to describe this… “pre emptied tank approach” or something similar might be more digestable… this needs to be considered throughout paper

122 – Q this is a conceptual model, not a comparison using data?

173 – sentence difficult to interpret

192 – “the” missing? Before decision

222 – should we term the bursts a hyetograph?

257 – can we plot these as line graphs? Should they not be scatter / bar charts?

274 – I am not sure if this chart is an appropriate format to include? Difficult to interpret.

349 – green lineis “total inflow” that would imply cumulative? It should eb simply “inflow rate”

393 – AEP missing?

414 –traditional again…

426 - …need to add something like “but it is acknowledged that taking a short time step (<1m) and having fully open valves for a short duration can also achieve a similar outcome in practical terms

429 – it would be good to comment or reference on the validity of these assumptions… to give authors a feel for how far from reality the concept is?

455 – again not sure this is the best format o display the data here…

Conclusions – something referring to the opportunity to deploy and test these strategies a one or more test beds is worth adding here? To establish how realworld performance varies from hypothetical?

References – this generally looks globally at the research literature but the authors could double check Water Journal papers that cover RWH / smart stormwater to ensure other relevant references are not missed. Some European and USA papers might be worth seeking out and touching on in the lit review elements. e.g. Ward, Campisano, Hunt, Gee, 

556 – reference undated?
